# SERPIN-Derived Small Peptide (SP16) as a Potential Therapeutic Agent against HIV-Induced Inflammatory Molecules and Viral Replication in Cells of the Central Nervous System

**DOI:** 10.3390/cells12040632

**Published:** 2023-02-15

**Authors:** Yemmy Soler, Myosotys Rodriguez, Dana Austin, Cyrille Gineste, Cohava Gelber, Nazira El-Hage

**Affiliations:** 1Department of Immunology and Nanomedicine, Herbert Wertheim College of Medicine, Miami, FL 33199, USA; 2Department of Chemistry and Biochemistry, Florida International University, Miami, FL 33199, USA; 3Serpin Pharma, 9501 Discovery Blvd Suite 120, Manassas, VA 20109, USA

**Keywords:** serine protease inhibitors, blood-brain barrier, low-density lipoprotein receptor related-protein 1, therapeutic efficacy, intranasal delivery

## Abstract

Despite the success of combined antiretroviral therapy (cART) increasing the survival rate in human immunodeficiency virus (HIV) patients, low levels of viremia persist in the brain of patients leading to glia (microglia and astrocytes)-induced neuroinflammation and consequently, the reactivation of HIV and neuronal injury. Here, we tested the therapeutic efficacy of a Low-Density Lipoprotein Receptor-Related Protein 1 (LRP-1) agonistic small peptide drug (SP16) in attenuating HIV replication and the secretion of inflammatory molecules in brain reservoirs. SP16 was developed by Serpin Pharma and is derived from the pentapeptide sequence of the serine protease inhibitor alpha-1-antitrypsin (A1AT). The SP16 peptide sequence was subsequently modified to improve the stability, bioavailability, efficacy, and binding to LRP-1; a scavenger regulatory receptor that internalizes ligands to induce anti-viral, anti-inflammatory, and pro-survival signals. Using glial cells infected with HIV, we showed that: (i) SP16 attenuated viral-induced secretion of pro-inflammatory molecules; and (ii) SP16 attenuated viral replication. Using an artificial 3D blood-brain barrier (BBB) system, we showed that: (i) SP16 was transported across the BBB; and (ii) restored the permeability of the BBB compromised by HIV. Mechanistically, we showed that SP16 interaction with LRP-1 and binding lead to: (i) down-regulation in the expression levels of nuclear factor-kappa beta (NF-κB); and (ii) up-regulation in the expression levels of Akt. Using an in vivo mouse model, we showed that SP16 was transported across the BBB after intranasal delivery, while animals infected with EcoHIV undergo a reduction in (i) viral replication and (ii) viral secreted inflammatory molecules, after exposure to SP16 and antiretrovirals. Overall, these studies confirm a therapeutic response of SP16 against HIV-associated inflammatory effects in the brain.

## 1. Introduction

Microglia/macrophages and, to a lesser extent, astrocytes maintain low levels of human immunodeficiency virus (HIV) infection and become virus reservoirs that impede viral eradication in the central nervous system (CNS) [1,2]. Despite the great success of combined antiretroviral therapy (cART) in combating viral infection and increasing the longevity of people living with HIV (PLWH), cART does not directly target the inflammatory cascades secreted by glia. This cascade is believed to be the primary cause of neuronal injury and dysfunction related to HIV-associated pathology [3,4,5]. PLWH still suffer from chronic inflammation that can lead to death due to non-related AIDS events [6,7]. Even though concentrations of several antiretroviral drugs have been detected in postmortem brain tissues of HIV subjects, hidden virus in brain reservoirs is still a significant hurdle toward a functional cure, and a definite HIV cure has not been achieved, yet. Moreover, high concentrations and life-long use of cART in the brain correlated with worsened neurocognitive performance, showing that cART can be neurotoxic [8] and can be associated with the development of neurocognitive disorder associated with HIV [9,10,11]. Considering these and other limitations related to the severe side effects, toxicities, drug-drug interactions, and compliance problems with cART, many alternative drugs with a combined anti-inflammatory response that target the brain reservoirs directly or indirectly are under investigation [12,13,14,15].

HIV circulates throughout the bloodstream and enters the CNS during the first weeks of infection, mainly mediated by infected monocytes and CD4^+^ T lymphocytes. These cells are generally attracted to inflammation sites and can enter the perivascular spaces [16]. Monocytes are often described as the “Trojan Horse” in CNS viral entry [17,18]. HIV can also reach the CNS through the lymphocytes, which are capable of harboring viruses that can reproduce in macrophages [19]. Furthermore, neuro-invasion may also happen by circulating virus crossing the BBB intracellularly through endothelial cells, especially when the permeability of this layer is compromised [16]. When the virus reaches the brain by either one of these pathways, it infects and activates microglia. These cells can also be activated by viral proteins such as gp120 and Tat released from infected cells. The activated microglia then release a list of neurotoxins, such as arachidonic acid, quinolinic acid, glutamate, L-cysteine, platelet-activating factor (PAF), and free radicals. These neurotoxic substances induce synaptic damage and neuronal injury and contribute to astrocyte activation. Activated astrocytes release glutamate, free radicals, and other neurotoxic substances that induce metalloproteinases, increase calcium influx, and further aggravate neuronal damage [20]. This creates an ideal environment for viral replication, chronic inflammation, neurotoxicity, and ultimately neurodegeneration [21].

Serpins are the largest and most broadly distributed superfamily of serine protease inhibitors. They inhibit plasma serine proteases such as trypsin, elastase, thrombin, and others [22]. Several studies have elucidated the role of serpins in the brain, including its anti-inflammatory, anti-viral, pro-survival, and neuro-protective effects [23,24]. Serpins have two essential features: a reactive center loop (RCL) that is responsible for the stabilization of the protein, as well as protease recognition and inhibition, and a short peptide motif (five to 11 amino acids) that is exposed when serpins undergo a conformational change to bind to serine proteases to inactivate them [24,25,26]. This short peptide motif is responsible for binding to low-density lipoprotein receptor-related protein-1 (LRP-1) [27], a large ubiquitous endocytic receptor expressed in the plasma membrane of many cells in several tissues, functioning as a scavenger, scaffold, or signaling receptor [28,29,30]. LRP-1 regulates many important functions, including lipoprotein and lipid metabolism, receptor-mediated endocytosis, protease degradation, apoptosis and autophagy, BBB permeability, inflammation, and cell growth and survival [31,32,33]. Multiple studies have confirmed the expression of this receptor in neurons, glia (astrocytes and microglia), and endothelial cells [34,35,36].

Small peptide 16 (SP16) is a short linear synthetic molecule developed by Serpin Pharma for treating autoimmune and inflammatory diseases [25,29,31]. SP16 contains the 5-amino acids sequence derived from the endogenous serine protease inhibitor, α-1-antitrypsin (A1AT), responsible for binding to LRP-1 [25]. A1AT has been widely researched and safely used in the clinic for over 30 years. Due to the high sequence homology of SP16 with the endogenous serpin, A1AT, SP16 is likely to be non-immunogenic [37,38] with anti-inflammatory and immune-modulatory effects. Here we explored the anti-inflammatory therapeutic potential of SP16 in brain cells with possible anti-viral response against HIV infection.

## 2. Materials and Methods

### 2.1. Human Primary Cells and Cell Lines

Human primary cells obtained from ScienCell Research Laboratories (Carlsbad, CA, USA) included primary astrocytes (catalog # 1800), primary brain microvascular endothelial cells (BMECs) (catalog # 1710), and primary pericytes (catalog # 1200). Human cell lines obtained from American Type Culture Collection (ATCC; Manassas, VA, USA) included microglia (catalog # CRL-3304). Cells were seeded in 24-well plates coated with poly-L-lysine (0.1 mg/mL) and maintained for 5–10 days at 37 °C and 5% CO_2_ [39,40,41,42,43]. Cells were maintained in their respective growth medium supplemented with fetal bovine serum-FBS (10%) procured from Hyclone (Logan, UT, USA), and penicillin/streptomycin (1%) procured from Invitrogen (Waltham, MA, USA). Media was discarded every other day, and fresh media was added.

### 2.2. HIV Infection

The HIV-1_SF162_ strain, from the macrophage lineage isolated from Dr. Jay Levy, was obtained through the NIH (Germantown, MD, USA) Research and Reference Reagent Program. Cells were infected with 1 ng/mL of HIV (p24/10^6^ cells) as performed previously [39,43,44]. After 24 h of exposure to the virus, media was discarded, cells were rinsed with PBS, and fresh media was added for another 24 h [40,44].

### 2.3. SP16 Treatment

Both SP16 and Cy5.5 labeled SP16 were obtained from Serpin Pharma (Manassas, VA, USA). SP16 obtained in powder form was dissolved in distilled water to a concentration of 3 mg/mL as per the manufacturer’s protocol. Human brain cells were treated at different concentrations of SP16 (0, 1, 25, 100, 150 μg/mL) and at different time points (6, 12, 24, 48, and up to 96 h), depending on the experiments performed.

### 2.4. In Vitro 3D Blood Brain Barrier (BBB) Model

Human primary endothelial cells were grown to confluence on the upper (luminal) surface of a 3 μm transwell membrane insert and astrocytes and pericytes were grown to confluency on the bottom (apical) surface of the inserts close to the endothelial cells to simulate the structure of the BBB. Transwell inserts were submerged in their respective media, as described previously [40]. The BBB was cultured for approximately 5 days and the inserts were transferred into a 24-well plate containing both HIV-infected and non-infected microglia seeded at the bottom of each well.

### 2.5. MTT Assay

Cell proliferation and survival were measured using an MTT (3-[4,5-dimethylthiazol-2-yl]-2,5 diphenyl tetrazolium bromide) cell growth assay kit from MilliporeSigma (Temecula, CA, USA). Human brain cells at a cell density of 3 × 10^4^ cells/well, were exposed to SP16 at different concentrations. After 24 and 96 h, 10 μL of 12 mM MTT solution was added to each well. After 4 h of incubation, DMSO was added to dissolve the formazan crystals. The absorbance in each well was read at 570 nm with a reference wavelength of 630 nm on a Synergy HTX microplate reader from BioTek (Winooski, VT, USA).

### 2.6. Trypan Blue Staining Assay

Cell viability was measured using trypan blue exclusion assay, which allows for the identification of live (unstained) and dead (blue) cells. Human brain cells were grown to confluency in a 24-well plate and exposed to SP16 at increasing concentrations. After 24 and 96 h, cells were harvested using enzyme-free cell dissociation buffer followed by centrifugation. Pellets were re-suspended in their specific media, and 10 μL of suspensions were diluted in an equal volume of 0.40% trypan blue dye from Bio-Rad (catalog #145-0013, Hercules, CA, USA). 10 µL was loaded on a counting chamber, and a TC20 automated cell counter determined the percentage of live cells from Bio-Rad (Hercules, CA, USA). Cell viability was expressed as the percentage of live cells out of total cells.

### 2.7. Enzyme-Linked Immunosorbent Assay (ELISA)

Levels of cytokines and chemokines in cell supernatant and postmortem brain homogenates were measured using an ELISA kit from R&D Systems (Minneapolis, MN, USA) as per the manufacturer’s protocol. Viral load (HIV and EcoHIV) was confirmed by a p24 ELISA kit from ZeptoMetrix (Buffalo, NY, USA) as performed previously [39,43,44]. The optical density was read at A_450_ with wavelength correction at A_570_ on a Synergy HTX plate reader from BioTek (Winooski, VT, USA).

### 2.8. Western Blotting

Proteins in brain cell lysates and postmortem brain homogenates were separated by SDS-PAGE, transferred to PVDF membranes, and subsequently incubated with primary antibodies from Santa Cruz Biotechnology (Dallas, TX, USA), including NF-κB at 1:200 dilution (catalog # sc-8008) and β-actin at 1:200 dilution (catalog # sc-47778). Antibodies from Abcam (Boston, MA, USA), including Akt at dilution 1:500 (catalog # ab-38449) and LRP-1 at dilution 1:50,000 (catalog # ab-92544). After primary antibodies, membranes were incubated with horseradish peroxidase-conjugated secondary antibodies followed by exposure to SuperSignal West Femto Substrate from Thermo Scientific (Waltham, MA, USA). Membranes were visualized using a ChemiDoc imaging system from Bio-Rad (Hercules, CA, USA) and protein expression was calculated using Image J software from NIH (Bethesda, MD, USA).

### 2.9. Matrix-Assisted Laser Desorption/Ionization-Time of Flight (MALDI-TOF) Mass Spectrometry (MS)

Samples were cleaned with C18 Zip Tips for the extraction of peptides. Zip Tip cleaning was performed using Millipore’s manual with utilizing 100% Acetonitrile (ACN) as the wetting solution, 0.1% TFA as the equilibration and wash solutions, and 0.1% TFA/50% ACN as the elution solution. After Zip Tip extraction, the extracted aliquots were placed in clean Eppendorf tubes for subsequent co-mixing with the α- Cyano-4-hydroxycinnamic acid (CHCA) matrix for analysis. MALDI-TOF-MS analysis was conducted on the Bruker AutoFlex III instrument, utilizing the dried droplet technique. The mass spectrometer was operated in reflection mode and in the positive (+) ion polarity. Samples were prepared by mixing a 6 μL droplet of a saturated solution of the α-Cyano-4-hydroxycinnamic acid (CHCA) with a 6 μL droplet of the Zip Tip extract in a plastic Eppendorf tube. The matrix solvent system was composed of acetonitrile:water (50:50 ratio) with 0.5% TFA (trifluoroacetic acid). Thereafter, a 1 μL droplet of the mixed sample solution was deposited on the stainless steel MALDI plate and allowed to air dry. Additionally, an extra 1 μL droplet of matrix was added to sample time point 150 mg 4 h to increase homogeneity in the crystalized surface. Subsequently, the MALDI plate was inserted into the MALDI-MS for analysis. The MALDI-TOF instrument was calibrated with the AnaSpec PepMix Calibration solution, utilizing 6 calibration points ranging from 900 to 3600 Da with a mass error <10 ppm.

### 2.10. Trans-Endothelial Electrical Resistance (TEER)

Electrical resistance across the endothelial monolayer formed in the transwell of the in vitro BBB was measured using a Millicell ERS microelectrode from Millipore (Bedford, MA, USA) as performed previously [45]. The resulting TEER values were calculated minus the blank media control.

### 2.11. Fluorescein Isothiocyanate (FITC) Dextran Trans-Epithelial Permeability Assay

BBB transwell inserts were exposed to different concentrations of SP16 (0, 1, 100, and 150 μg/mL). After 12 h, media containing the treatment from the upper chamber of each transwell insert was discarded and the cell inserts were transferred into a new 24-well plate with 750 μL of fresh media. 150 μL of FITC-dextran was added to the upper chamber of the insert and the plate was incubated for 2 h in the dark at 37 °C. Samples were collected from the bottom chamber after 2 h and fluorescence intensity was measured at excitation wavelength 485 nm and emission wavelength 520 nm using plate reader instrument (Bio Tek). FITC-dextran permeability across the inserts was calculated using a standard curve based on serial dilution from the stock of FITC-dextran (100–0.098 μg/mL).

### 2.12. Flow Cytometry

Primary human astrocytes and microglia were fixed in staining buffer containing 2% BSA, 4% formaldehyde, washed, and incubated with anti-LRP-1 (catalog # ab-92544) from Abcam (Boston, MA, USA), anti-GFAP (catalog # MAB360) from Millipore (Bedford, MA, USA), and anti-Iba 1 (catalog # 019-19741) from Wako Chemicals (Richmond, VA, USA) for 1 h at 4 °C. After washing, cells were incubated in 2.5 uL secondary antibody for 20 min at 4 °C. Cells were washed and resuspended in staining buffer for analysis on an Accuri C6 flow cytometer from BD Biosciences (San Jose, CA, USA).

### 2.13. Immunocytochemistry

Human brain cells and postmortem brain tissues on culture slides (Thermo Scientific, Waltham, MA, USA) were fixed in 4% paraformaldehyde, permeabilized with 0.1% Triton X-100, blocked in 10% milk/0.1% goat serum, and immunolabeled. Primary antibodies from Thermo Fisher Scientific (Waltham, MA, USA) included Occludin conjugated to the Alexa Fluor 488 dye at 1:1000 dilution (catalog # 331588), from Abcam (Boston, MA, USA) included LRP-1 at 1:100 dilution (catalog # ab-92544), from Millipore (Burlington, MA, USA) included GFAP at 1:400 dilution (catalog # MAB360), and MAP-2 at 1:100 dilution (catalog # MAB378), and from Wako chemicals (Richmond, VA, USA) included Iba-1 at 1:50 dilution (catalog # 019-19741). A secondary antibody from Molecular Probes (Carlsbad, CA, USA) was used to visualize immunoreactivity and 4′,6-diamidino-2-phenylindole (DAPI) from Thermo Fisher Scientific (Waltham, MA, USA) was used to label cell nuclei. Cells were imaged using an inverted fluorescence microscope (Zeiss, Germany).

### 2.14. Gene Silencing

Human primary astrocytes and endothelial cells seeded in 24-well plates at 80% confluency were transfected overnight with small interfering (si) RNA against LRP-1, using Lipofectamine 2000 (Invitrogen, Carlsbad, CA, USA) in Opti-MEM media according to the manufacturer’s protocol. After overnight incubation, Opti-MEM media was replaced with corresponding culture media. After 48 and 96 h post transfection, cells were treated with SP16 (25 and 100 μg/mL) for 12 and 24 h.

### 2.15. Animal Use

C57BL/6J mice (Jackson Laboratory, Bay Harbor, ME, USA) were bred in the animal facility at Florida International University. Mice were placed 3–5 per cage and maintained in a controlled temperature and humidity room. The animals were cared for in accordance with the Principles of Animal Care outlined by the National Institutes of Health and approved by the Institutional Animal Care and Use Committee of Florida International University. Protocol Approval: IACUC-21-007-CR01. The EcoHIV-NDK plasmid used was provided by Dr. David Volsky at Mount Sinai school of medicine. The plasmid was transfected in HEK293T cells to generate EcoHIV stocks as previously described [46]. For the in vivo experiments, male mice between the ages of 4–6 months were administrated 1 mL of either saline or 1 × 10^6^ pg p24 units of EcoHIV by intraperitoneal (IP) cavity. After 3–4 weeks, animals received cART daily via IP administration. The cART regimen consisted of Emtricitabine, Ritonavir, and Atazanavir at a concentration of 100 mg/kg each, based on previously published literature by others [12,47,48,49]. A subset of cART treated animals received SP16 every other day at concentration of 10 μg/kg, via the intranasal route, as described previously by us [41]. After treatments (at day 5), mice were sacrificed and whole brains, excluding the brainstem and the olfactory bulb, were collected. Half of the recovered brain hemisphere was used for immunohistochemistry and LC-MS/MS while the other half was homogenized and used for biochemical assays. For protein biochemical assays, postmortem brains were homogenized in cell lysis buffer (Thermo Scientific, Waltham, MA, USA) using a bead-beater apparatus (MagNA Lyser, Roche, Indianapolis). Bioavailability of SP16 in the brain was monitored with LC-MS/MS using 703 (triple charged) parent ion and its two daughter ions 129.1 and 84.1. For the in vitro murine experiments, primary mixed glia cultures composed of astrocytes (~80% GFAP immune-positive cells) and microglia (~20% Iba-1 immuno-positive cells) were recovered from midbrains of C57BL/6J pups scarified according to IACUC guidelines as described previously [50,51]. Cells were seeded in 24-well plates coated with poly-L-lysine (0.1 mg/mL) and maintained for 5–10 days at 37 °C and 5% CO_2_ [39,40,41,42]_._ Cells were maintained in their respective growth medium, supplemented with glucose (2 mg/mL) procured from Sigma Aldrich (St. Louis, MO, USA), Na_2_HCO_3_ (6 mM) procured from Invitrogen (Waltham, MA, USA), fetal bovine serum-FBS (10%) procured from Hyclone (Logan, UT, USA), and penicillin/streptomycin (1%) procured from Invitrogen (Waltham, MA, USA). Cells were infected with EcoHIV at 1 ng/mL of HIV (p24/10^6^ cells) as performed previously. After 24 h of exposure to the virus, media was discarded, cells were rinsed with PBS, and fresh media was added for another 24 h.

### 2.16. Statistical Analysis

Results are reported as the mean ± SEM of 3 independent experiments. Data were analyzed using one or two-way ANOVA analysis followed by Dunnett multiple comparisons (GraphPad Prism8 software, La Jolla, CA, USA). A value of *p* < 0.05 was considered significant.

## 3. Results

### 3.1. SP16 Causes Both Anti-Inflammatory and Anti-Viral Responses in Microglia Infected with and without HIV Infection

The function of SP16 as both an anti-inflammatory and anti-viral agent was measured in microglia seeded in wells with a BBB insert (Figure 1A,C,E,G,I) and in wells without inserts (Figure 1B,D,F,H,J). After 12 h post-exposure with SP16 at increasing concentrations (1 and 100 µg/mL), the collected supernatant was used to measure the levels of inflammatory molecules including interleukin (IL-6 and IL-8), and monocyte chemoattractant protein (MCP-1) by ELISA [39,43,44]. As shown in Figure 1C,D, the secretion of MCP-1 was significantly decreased by more than 50% in both non-infected (gray bars) and HIV-infected (black bars) microglia that were seeded in plates with and without BBB. The decrease in MCP-1 appeared to be concentration dependent. Secretion of IL-6 and IL-8 was decreased in supernatant recovered from microglia without the BBB insert (Figure 1F,H) when compared to microglia seeded in plates with the BBB (Figure 1E,G). Likewise, viral load, measured by p24 gag protein ELISA was attenuated in a concentration-dependent manner by SP16 in microglia cells plated in wells without the BBB (Figure 1J). Overall, findings show that SP16 is an effective anti-inflammatory agent with anti-viral properties. Interestingly, the results with the BBB also suggest that the SP16 might not be transported effectively. To eliminate any doubts about the possibility of SP16 movement, the artificial BBB was used to measure peptide transportation.

### 3.2. SP16 Is Transported across an Artificial Blood-Brain Barrier (BBB)

Since we were interested in the function of SP16 in brain cells, it was imperative to investigate SP16′s ability to penetrate the BBB. SP16 was added to the apical side of the BBB and after 6 and 12 h, the bottom (basal level) media was collected and used for MALDI-TOF-MS analysis (Figure 2). The mass spectra show the analysis of the SP16 samples. Mass spectrometry analysis demonstrate an ion at m/z 2106.2 detected in samples used at the following concentrations: 100 μg/mL after 6 h, 150 μg/mL after 6 h (Figure 2A), at 25, 100 and 150 μg/mL concentrations after 12 h (Figure 2B). The isotope pattern observed for the ions at m/z 2106.2-2108.2 (arrows) agrees with the isotopic pattern provided by the Serpin Pharma data sheet and the stock solution of SP16 (Serpin standard; C102H160N24O24). In addition, the sodium adduct [M+H]+ of the Serpin peptide was detected at m/z 2128.2 (arrow). Although the specific concentration was not measured, findings using MALDI-TOF-MS demonstrate that SP16 (peptides) crossed the BBB, after incubation with 100 μg/mL, as early as 6 h. Next, we measured the resistance of the endothelial monolayer using TEER to quantitatively measure the integrity of tight junction dynamics in the BBB (Figure 2C–H). SP16 at concentrations of 1 and 100 μg/mL, were added to the apical side of the BBB up to 48 h. TEER values were acquired after treatment using a Millicell ERS microelectrode (Millipore, Bedford, MA, USA). Findings show an overall high resistance TEER value (or low permeability) of the tight junctions in the BBB. This indicates that SP16 does not compromise the integrity of the monolayer as the TEER values remained constant at about 300 Ωxcm^2^ for up to 48 h. In a parallel experiment, SP16 at concentrations of 25, 100 and 150 μg/mL, were added to the apical side of the BBB up to 12 h. FITC-dextran permeability assay was used to measure BBB integrity. The data in Figure 2I,J) shows increased FITC values (µg/mL), indicative of a high permeability in the tight junction of the BBB when cells are infected with HIV compared to low FITC values in cells not infected with HIV. More importantly, SP16 was able to lower the permeability of the monolayer starting at 100 μg/mL in HIV-infected cells. Overall, findings confirmed transportation of the peptide across the artificial BBB, although the exact concentration of SP16 that crossed was not determined. Furthermore, data demonstrate that SP16 does not affect BBB integrity but rather restored integrity when added to HIV-infected cells.

### 3.3. SP16 Induces Both Anti-Inflammatory and Anti-Viral Responses in Astrocytes and Endothelial Cells Infected with and without HIV Infection

Based on the microglia data, we further explored the efficacy of SP16 in other brain cells. As shown in Figure 3, the secretion of MCP-1 in primary human astrocytes was significantly attenuated by more than 90% in a concentration-dependent manner after 6 (Figure 3A,B) and 12 h (Figure 3C,D) post-treatment with SP16. Interestingly, HIV-induced MCP-1 was reduced by 65% after 6 h and by 63% after 12 h post-treatment with 1 μg/mL SP16. Attenuation of HIV-induced IL-6 (Figure 3F) and IL-8 (Figure 3H) were also detected after 12 h post-treatment with SP16 at concentrations at 1 and 25 μg/mL, respectively. The same supernatant was used to measure viral titers by HIV p24 gag protein ELISA. After 6 (Figure 3I) and 12 h (Figure 3J) post-treatment with SP16, viral titer detected by HIV p24 gag protein ELISA showed about 50% reduction at 25 μg/mL. Next, we investigated the anti-inflammatory effects of SP16 in primary endothelial cells exposed to increasing concentrations of SP16 (25 and 250 μg/mL). After 12 h post-treatment, the supernatant was collected, and the levels of inflammatory molecules and viral titer were measured by ELISA [39,43,44]. Likewise, after 12 h, SP16 significantly decreased the secretion of both MCP-1 by about 40% (Figure 3L) and IL-8 by about 15% (Figure 3N) in HIV-infected endothelial cells. Attenuation in the levels of HIV p24 (Figure 3O) was also detected 12 h post-treatment at 250 μg/mL SP16.

Toxicity with SP16 was further measured with an MTT (Figure 4A,C,E,G,I,K) and a Trypan Blue Staining (Figure 4B,D,F,H,J,L) assay, using a range of SP16 concentrations and two time points. As demonstrated in Figure 4, SP16 did not decrease cell viability in microglia (Figure 4A–D), astrocytes (Figure 4E–H), or endothelial cells (Figure 4I–L). Both assays showed minimal cell loss after SP16 treatment, indicating SP16 is non-toxic to brain cells. Overall, findings suggest that SP16 is an effective anti-inflammatory agent with potential anti-viral properties that could serve as a novel therapy for the treatment against HIV-induced pathology in the CNS.

### 3.4. SP16 Increases the Exression Level of LRP-1 and Akt Signaling in Brain Cells Infected with and without HIV Infection

Protein expression levels of LRP-1 were quantitively measured by flow cytometry in human microglia (Figure 5A) and astrocytes (Figure 5B). Among the microglia population gated (dotted circle) approximately 20% expressed the LRP-1 receptor and about 86% of astrocytes expressed the LRP-1 receptor. The expression of LRP-1 was also detected in endothelial cells using immunofluorescence (IF) staining (Figure 5G,H). In addition, expression levels of LRP-1 in the presence of SP16 in HIV-infected and non-infected human primary astrocytes (Figure 5C,D) and endothelial cells (Figure 5E,F) were measured by Western blotting after 12 h. As expected, the expression of LRP-1 was higher in HIV-infected cells (Figure 5D,F) compared to non-infected cells (Figure 5C,E). Our findings are consistent with published literature indicating LRP-1 expression increases during tissue injury [25,52]. Moreover, in both endothelial cells and astrocytes, reduced LRP-1 expression levels were detected with an increased concentration of SP16 (Figure 5C–F), which suggests receptor internalization or receptor proteolysis as a mechanism of action for lowering NF-κB and increasing Akt. To confirm our statement, we measured the expression levels of NF-κB and Akt in human primary astrocytes (Figure 6A–D) and endothelial cells (Figure 6E–H) exposed to 25 μg/mL of SP16. After 12 h post-treatment, cell lysates were collected, and protein expression levels were measured by Western Blotting. The expression levels of Akt were upregulated at 25 μg/mL of SP16 (Figure 6C,D,G,H), in both infected and non-infected human primary astrocytes and endothelial cells. However, at similar concentrations no significant changes in the expression level of NF-κB (Figure 6A,B,E,F) were detected. Of note, significantly reduced expression levels of NF-κB were detected in cells exposed to 100 (Figure 7) and 250 μg/mL of SP16, however, the expression level of β-actin (internal control) at 250 μg/mL SP16 was also reduced (data not shown). Based on our findings, we investigated whether the secretion of inflammatory molecules in brain cells could be regulated via SP16 binding to LRP-1, the Akt and the NF-κB signaling pathways.

### 3.5. LRP-1 Mediates SP16 Anti-Inflammatory Action in Human Brain Cells Via the NF-κB and Akt Signaling Pathways

Using a gene silencing approach, we confirmed the role of LRP-1 in the regulation of NF-κB and Akt expression in astrocytes (Figure 7A,C,E) and endothelial cells (Figure 7B,D,F). After 48 h post-transfection, astrocytes were exposed to 100 μg/mL SP16 and endothelial cells were exposed to 25 and 100 μg/mL SP16, for 12 h. As shown in astrocytes (Figure 7A; gray bar) and endothelial cells (Figure 7B; gray bar), silencing of the *LRP1* gene, caused a significant reduction in the expression levels of LRP-1. Reduced levels of LRP-1 expression were also detected in cells exposed to increased concentrations of SP16, which can be attributed to the binding of the peptide to the LRP-1 receptor and cell internalization and/or proteolytic cleavage of the receptor. Interestingly, the expression levels of NF-κB were significantly reduced in astrocytes exposed to 100 μg/mL of SP16 (Figure 7C; black bars), while silencing the *LRP-1* gene reverted the response in cells exposed with and without SP16 (Figure 7C; gray bars). On the contrary, silencing the *LRP-1* gene in astrocytes (Figure 7E; gray bar), caused a decrease in the expression level of Akt, that was partially reverted in cells exposed to SP16 (Figure 7E). In endothelial cells, gene silencing caused the levels of NF-κB to increase (Figure 7D; gray bars), and Akt to decrease (Figure 7F; gray bars). Interestingly, when *LRP-1* gene was silenced in the absence of SP16, expression levels of NF-κB remained increased and Akt remained decreased, although to a lesser extent as compared to cells exposed to SP16 without gene silencing. These finding confirm the importance of the LRP-1 receptor in regulating both the NF-κB and Akt signaling pathways in astrocytes and endothelial cells.

The cell supernatant collected was used to measure the secreted levels of pro-inflammatory molecules. Secreted levels of MCP-1, IL-6 and IL-8 were significantly decreased in astrocytes exposed to 100 µg/mL SP16 (Figure 8A–C; black bar). Likewise, secreted levels of IL-6 and IL-8 in endothelial cells (Figure 8E,F; black bars) showed a significant decrease after exposure to 100 µg/mL SP16. However, silencing the *LRP-1* gene caused a reversal in the secretion levels of MCP-1, IL-6, and IL-8 in astrocytes (Figure 8A–C; gray bars) and in the secretion of IL-6 and IL-8 in endothelial cells (Figure 8E,F; gray bars). Our results confirm that the therapeutic effects of SP16 are likely mediated by the interaction between the peptide and LRP-1. Altogether, our results agree with previously published data indicating that the expression of LRP-1 increases during tissue injury, leading to a down-regulation of the pro-inflammatory process through the NF-κB signaling pathway while promoting cell survival through the Akt signaling pathway [25,52].

### 3.6. SP16 Induces Therapeutic Responses in Murine Brains Infected with EcoHIV

To confirm our in vitro data, we investigated the anti-inflammatory and anti-viral responses of SP16 using mouse brain glia and in vivo infected-mouse model system. As this experiment was exploratory in nature, we used a conventional mouse model and the chimeric virus, EcoHIV, specifically the NDK strain, as it infects mouse brain cells with more efficiency than EcoHIV [46]. Glia recovered from mouse brain, infected with and without EcoHIV were exposed to SP16. After 12 h post-treatment, the supernatant was collected and used to measure the levels of secreted inflammatory molecules and viral titer by ELISA [39,43,44]. SP16 at concentration of 250 μg/mL significantly decreased the secretion levels of MCP-1 by about 80% (Figure 9A) and IL-6 by about 25% (Figure 9B). Viral replication measured by p24 gag protein was reduced by 30% (Figure 9C). Moreover, the expression levels of both LRP-1 and NF-κB, measured by immunoblotting, were significantly downregulated after exposure to 25 and 250 μg/mL of SP16, (Figure 9D,E) while expression of Akt was significantly induced in glia exposed to 250 μg/mL of SP16 (Figure 9F). In fact, NF-κB expression level was reduced by more than 75% when compared to infected glia without SP16 exposure (Figure 9E; 250 μg/mL of SP16). Overall, our results agree with our in vitro data and show increased expression of LRP-1 during cell injury, a decreased expression of NF-κB and an increased expression of Akt [25,52]. Lastly, adult male C57BL/6J mice were injected with EcoHIV via intraperitoneal administration. After 21 days animals received cART alone or in combination with SP16 (as illustrated in schematic diagram G). After 24 h mice were sacrificed, and postmortem brains recovered were homogenized to measure secreted inflammatory molecules and viral replication by ELISA. Infection with EcoHIV caused a significant increase in the secretion of MCP-1 (Figure 9H), RANTES (Figure 9I), and IL-6 (Figure 9J) by more than 50% as compared to brains exposed to PBS control. Infected brains exposed to cART alone showed a slight albeit significant decrease in the secretion of cytokines, while the addition of SP16, caused a staggering decrease of 48% in MCP-1 as compared to cART alone (Figure 9H) and a 35% decrease in RANTES as compared to cART alone (Figure 9I). An increased level of the p24 gag protein was detected, and this was about a 3.3-fold higher when compared to brains exposed to PBS control (Figure 9K). Exposure to cART caused a 28% decrease in p24 gag, that was further decreased by 19% after exposure with SP16 (Figure 9 K). Expression levels of LRP-1 (Figure 9L), NF-κB (Figure 9M) and Akt (Figure 9N) were also measured by immunoblotting. Although minimal changes were detected in the expression of LRP-1, EcoHIV significantly increased the expression levels of NF-κB (Figure 9L) while significantly decreased the expression levels of Akt (Figure 9M) when compared to control (PBS)-treated brains. Treatment with cART alone and in combination with SP16, lead to significant increase in LRP-1, a significant decrease in the protein expression level of NF-κB and an increase in the expression of Akt. In fact, treatment with combination SP16 and cART caused about 86% decrease in NF-κB as compared to cART alone, and about 18% increase in Akt as compared to cART (Figure 9M,N). Bio availability of SP16 peptide to the brain after intranasal delivery was detected with LC/MS-MS (Appendix A) while LRP-1 expression in brain tissues was observed using immunocytochemistry labeling (Appendix A).

## 4. Discussion

For the study described here, we used SP16, a peptide derived from the serine proteinase inhibitor (serpin), A1AT, found in abundance in human plasma. A1AT drugs have both pro- and anti-inflammatory structural regions, and when used to treat inflammation, the pro-inflammatory regions dampen the anti-inflammatory effect. SP16 contains only the anti-inflammatory region of A1AT, making the peptide 300 times more potent than A1AT. Moreover, the SP16 peptide does not need to bind to a serine protease to become active. In a phase 1 trial, the peptide showed no safety concerns [53], no off-target effects, no cardiac liability, no humoral immune response, and no drug-specific toxicity [25]. Findings presented here showed both an anti-inflammatory and to a lesser extent, an anti-viral potential of SP16 in brain cells, mediated via binding with LRP-1 and the NF-κB and Akt signaling pathways.

Our findings correlate with the anti-inflammatory effects of SP16 as previously published by Toldo et al. [25,37,54]. Their studies showed a reduction in the formation of the NLRP3 inflammasome in macrophages in vitro and a decrease in inflammation after acute myocardial infarction in a mouse model following SP16 treatment [25,29,31]. We demonstrate SP16 induces a significant anti-inflammatory response in both HIV-infected and non-infected microglia (Figure 1). We also showed that SP16 induces an anti-HIV response in viral-infected brain cells. Interestingly, both IL-6 and IL-8, along with viral-induced p24 levels, were unaffected by SP16 in cells with the BBB inserts compared to cells without the BBB inserts. Several reasons may account for this lack of anti-inflammatory response including peptide modification during interactions with the BBB. Additionally, receptor mediated endocytosis can lead to transcytosis but also degradation. However, to monitor SP16 trafficking across the BBB in real time, additional techniques, such as intravital two photon microscopy in mouse, are required, which is beyond the scope of this manuscript. Findings from Figure 1 bring up another important point, as we were able to detect peptide across the BBB, shown in Figure 2. To confirm the possibility of SP16 transportation, an artificial BBB was used to assess peptide transportation across the monolayer. Our findings in Figure 2 showed transport of SP16 across the BBB, which according to Wallet et al., is the major hurdle for an effective HIV treatment for the CNS [7]. In fact, we showed that SP16 penetrates the BBB at a dose of 100 μg/mL, and we also showed that even at low concentration (1 μg/mL), SP16 can decrease IL-6, suggesting that less than 1 μg/mL is crossing the BBB at 12 h. Moreover, peptide transportation did not disrupt the monolayer. This is an important finding since the integrity of the BBB plays a critical role in HIV treatment delivery into the brain [55]. Instead, we showed that SP16 restored the integrity of the BBB in cells infected by HIV (Figure 2).

We also assessed the efficacy of SP16 in astrocytes, as these cells can be infected by HIV and secrete inflammatory molecules. Although astrocytes do not infect to the same extent as microglia, they have been shown to support low levels of HIV infection and exhibit traits of latency [56]. We did not use the artificial BBB, as we noticed some discrepancies when using the transwell inserts. The secretion of MCP-1 was decreased in a concentration-dependent manner after 6 and 12 h, while IL-6 and IL-8 levels were decreased only after 12 h post-treatment (Figure 3). Viral protein p24 was also attenuated in astrocytes after 6 and 12 h of treatment. Since HIV and released viral proteins can lead to endothelial dysfunction, which causes BBB disruption, a common feature of neurodegeneration and neurocognitive disorders [57,58,59], we also tested the action of SP16 in brain microvascular endothelial cells (Figure 3), and showed similar results as with astrocytes. While we are not certain of the mechanism of how SP16 can attenuate viral titer (in the absence of cART), it is well recognized that the NF-κB transcription factor family members form distinct transcriptionally active homo- and heterodimeric complexes such as the NF-κB heterodimer p50/p65 [60], which is essential for the replication of HIV [61,62,63]. It has been shown that A1AT interacts with IkBα, which blocks the action of NF-κB, inhibiting HIV replication [64,65,66]. Indeed, our studies also showed a correlation between reduced expression of viral load with a reduced expression of NF-κB.

LRP-1 is abundantly expressed after injury in the CNS and other organs including the liver, kidney, lung, vasculatures, and in the heart [25,30,67]. In the CNS, this receptor has been found in several brain cells including microglia, astrocytes, endothelial cells, and neurons [34,68,69,70,71] using various techniques including hybridization assays, immunostaining, and electron microscopy [52]. The binding of SP16 to LRP-1 in mouse model with myocardial infarction and in a mouse model with lung pneumonia, resulted in an anti-inflammatory and pro-survival action. The pro-survival signaling was exerted through the action of protein kinase AKT [68,72], and the anti-inflammatory actions were mediated through suppression of the c-Jun N-terminal kinase (JNK) [73,74] and the NF-κB pathways [75]. Our findings correlate with previously reported publications. Here, we showed increased expression of LRP-1 after exposure with SP16, which correlated with an increase in Akt and a downregulation in the NF-κB expression levels. The reduced expression of NF-κB correlated with a decreased secretion in both cytokines and chemokines, after exposure with SP16 treatment (Figure 8).

LRP-1 has been shown to conduct many functions including the regulation of BBB permeability [76]. More importantly, this receptor mediates intracellular delivery to the CNS by endothelial transcytosis, an effective endogenous method for crossing the BBB [77,78]. For example, Nikolakopoulou et al. reported that an endothelial LRP-1 knockout caused loss of tight junctions, resulting in BBB impairments. This study concluded that the LRP-1 receptor is essential for clearing out toxins at the BBB and therefore, protecting against neurodegeneration [36]. Furthermore, LRP-1 has been shown to play a major role in the process of atherosclerosis by mediating endocytosis of LDL particles [79]. Endothelial LRP-1 has also been found to transport amyloid-β, which drives Alzheimer’s pathology, across the BBB to mediate its clearance [71]. More importantly, this receptor has been shown to be capable of mediating intracellular delivery of antibodies to the CNS with the use of polymersomes by transcytosis across the endothelial BBB [77]. According to Potere et al., LRP-1 acts in three different ways: as a scavenger, as signaling, or as scaffold receptor. As a scavenger receptor, LRP-1 internalizes a wide range of ligands including lipoproteins, growth factors, and protease inhibitor complexes [29]. As a signaling receptor, it regulates nuclear transcriptional signaling by generating a fragment of its intracellular domain (LRP-ICD) by proteolysis and translocating it from the cytoplasm to the nucleus to modulate gene transcription [29,31]. As a scaffold receptor, it serves as an anchor for different proteins and molecules coming from a co-receptor to eventually transmit the signal downstream [29,52]. Interestingly, in Figure 7, we measured a decrease expression in LRP-1 with increased concentrations of SP16, which we suspect can be attributed to receptor internalization or proteolysis. Although the specific mechanism that occurs when SP16 binds to LRP-1 remains unclear, we believe that it is one of the first two mentioned above. This would explain why less expression of LRP-1 was detected in cells exposed to higher concentrations of SP16. We believe that the receptor is either internalized with the ligand (SP16), or the receptor is cleaved, and the fragment generated is translocated. We found that silencing the LRP-1 receptor reverted the effects of SP16 on NF-κB and Akt expression levels (Figure 7). Moreover, silencing LRP-1 in the absence of SP16 increased NF-κB and decreased Akt expression levels, suggesting additional endogenous ligands could be involved in controlling these signaling pathways. We also showed that silencing LRP-1 attenuated the therapeutic characteristics of SP16 (Figure 8). The mechanism by which LRP-1 interacts with SP16 needs to be further investigated and explored.

The anti-inflammatory and anti-viral effects of SP16 was confirmed using the EcoHIV mouse model (Figure 9). Numerous studies have shown that EcoHIV is capable of infecting cells causing neurocognitive impairment, as well as transmitting the virus [80,81]. Furthermore, another study demonstrated an increase in inflammatory molecules leading to hippocampal dysfunction after intranasal administration of this virus [82]. The behavioral impairments caused by EcoHIV have also been investigated by others [83,84,85,86]. For drug delivery, we used the intranasal drug delivery method for the transport of SP16 to the brain in mice. Several studies have established the advantages of the intranasal drug delivery method. Gel formulations such as sumatriptan succinate, zolmitriptan, and rizatriptan for migraine, ropinirole for Parkinson’s, and venlafaxine for depression have shown to be successfully delivered by intranasal route [87,88,89,90]. Other formulations such as nanoparticles have been formulated as efficacious nasal delivery systems [91,92,93,94]. Nasal sprays have also been designed for intranasal delivery including dihydro ergotamine, sumatriptan, and zolmitriptan for migraine, and ketamine for depression [95,96,97,98]. Furthermore, the compounds mentioned above, such as zolmitriptan, have also been formulated as a dry powder to treat migraine, and ropinirole as a polymer micro-particle to treat Parkinson’s [99,100]. As shown in our infected animal studies, high levels of secreted MCP-1, IL-6, and IL-8 were detected in EcoHIV infected brain tissues that were only partially attenuated by cART alone but were significantly attenuated in brains combined with SP16; overall, supporting the dual therapeutic action of SP16 as an anti-inflammatory and anti-viral molecule.

There were some limitations to this study, which includes the use of human microglia cell lines (HMC3) instead of primary human microglia cells, which were no longer commercially available. Non-infected HMC3 cells produced the same amount of pro-inflammatory cytokines as HIV-infected cells (Figure 1), which does not occur in human astrocytes (Figure 3). Another limitation was the use of EcoHIV in the in vivo animal studies versus the use of HIV in the in vitro cell culture studies. We used EcoHIV, as it does not possess gp120, required for binding to human CCR5/CD4 receptors, making the viral strain safer to handle in live mouse studies. Despite the differences in EcoHIV viral genome when compared to regular HIV, our data showed similar findings. Lastly, viral load was measured by p24 gag ELISA, while a more quantitative approach such as LTR-primer specific RT-PCR would have been complimentary. Despite these limitations, the study presented here provides a proof-of-concept for the anti-inflammatory and anti-viral responses of SP16. Ongoing studies will further investigate the efficacy of SP16 in reversing neurological pathologies associated with HIV in the CNS, using humanized mouse model and non-human primate model.

## Figures and Tables

**Figure 1 cells-12-00632-f001:**
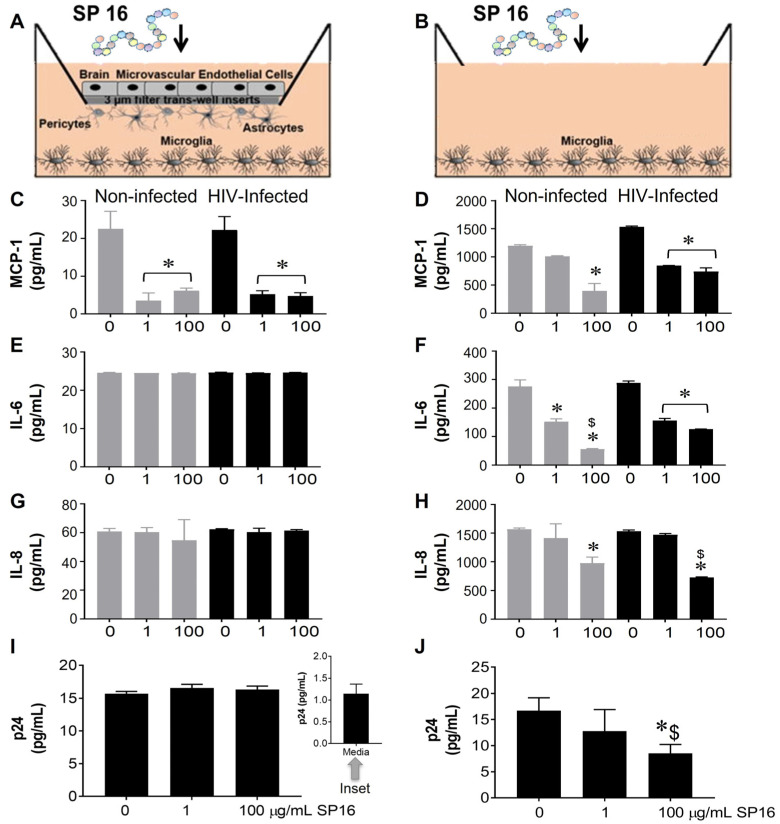
Anti-inflammatory and anti-viral responses of SP16 in microglia. SP16 at increasing concentrations was directly added to cells (**B**,**D**,**F**,**H**,**J**) or to the luminal surface of the BBB inserts (**A**,**C**,**E**,**G**,**I**). Supernatant was collected after 12 h and inflammatory molecules (**A**–**H**) and p24 (**I**,**J**) were measured by ELISA. Gray bars = non-infected cells, black bars = infected cells. Results are reported as the mean ± SEM of 3 independent experiments. Data were analyzed using one or two-way ANOVA analysis followed by Dunnett multiple comparisons. A value of *p* < 0.05 was considered significant. * vs. 0 SP16 (media); ^$^ vs. 1 μg/mL SP16.

**Figure 2 cells-12-00632-f002:**
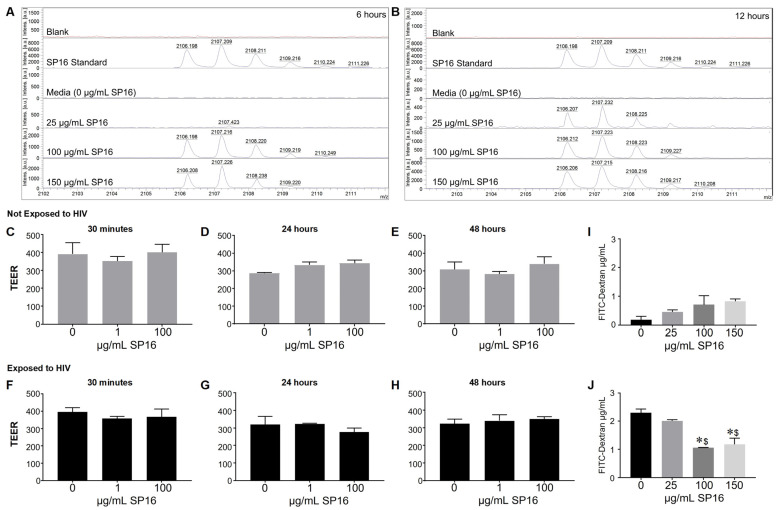
Movement of SP16 across an artificial blood-brain barrier (BBB). Supernatant was collected after 6 h (**A**) and 12 h (**B**). MALDI-TOF-MS analysis was conducted on the Burker AutoFlex III instrument to test the transportation of different SP16 concentrations across the BBB at the indicated time points. Integrity of the BBB was measured using a Millicell ERS microelectrode (**C**–**H**) and by a FITC-dextran Assay (**I**,**J**). Results are reported as the mean ± SEM of 3 independent experiments. Data was analyzed using one or two-way ANOVA analysis followed by Dunnett multiple comparisons A value of *p* < 0.05 was considered significant. * vs. media; ^$^ vs. 25 μg/mL SP16.

**Figure 3 cells-12-00632-f003:**
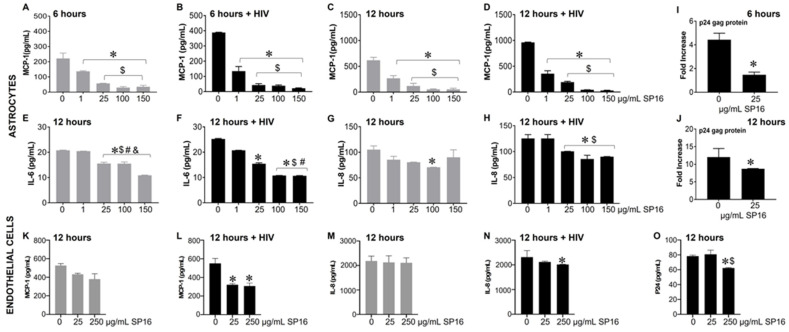
Anti-inflammatory and anti-viral responses of SP16 in astrocytes and endothelial cells. Supernatant was collected after 6 (astrocytes) and 12 (astrocytes and endothelial cells) hours and used to measure inflammatory molecules (**A**–**H**) in astrocytes, (**K**–**N**) in endothelial cells, and viral protein p24 (**I**,**J**) in astrocytes, (**O**) in endothelial cells, by ELISA. Gray bars = non-infected cells, black bars = infected cells. Results are reported as the mean ± SEM of 3 independent experiments. Data were analyzed using one or two-way ANOVA analysis followed by Dunnett multiple comparisons. A value of *p* < 0.05 was considered significant. * vs. media; ^$^ vs. 1 μg/mL SP16; ^#^ vs. 25 μg/mL SP16 (in astrocytes); ^$^ vs. 25 μg/mL SP16 (in endothelial cells); ^&^ vs. 100 μg/mL SP16.

**Figure 4 cells-12-00632-f004:**
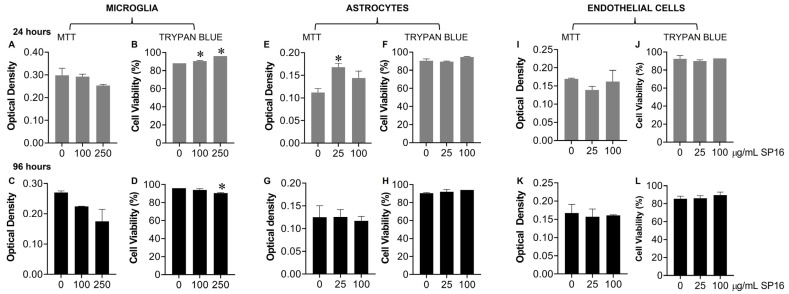
SP16 does not affect cell viability in microglia, astrocytes, or endothelial cells. SP16 at increasing concentrations was added to microglia (**A**–**D**), astrocytes (**E**–**H**), and endothelial cells (**I**–**L**). MTT (**A**,**C**,**E**,**G**,**I**,**K**); a Trypan Blue Staining (**B**,**D**,**F**,**H**,**J**,**L**) assay was used to assess cell viability at 24 h (gray bars) and 96 h (black bars). Results are reported as the mean ± SEM of three independent experiments. Data were analyzed using one or two-way ANOVA analysis followed by Dunnett multiple comparisons. A value of *p* < 0.05 was considered significant. * vs. media.

**Figure 5 cells-12-00632-f005:**
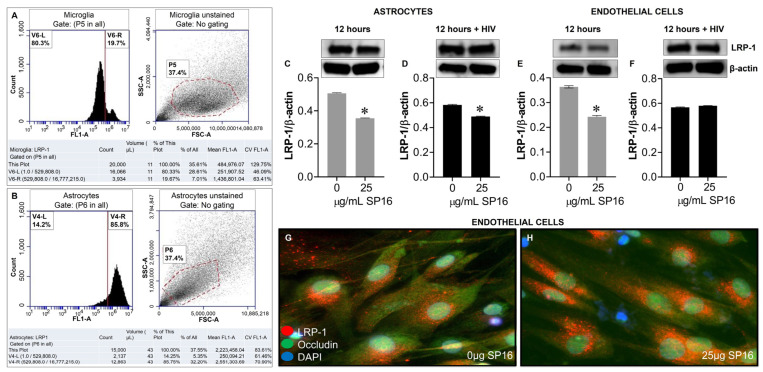
LRP-1 is expressed in brain cells. Protein expression levels of LRP-1 was quantitatively measured by flow cytometry in microglia (**A**) and astrocytes (**B**), and by immunoblotting in astrocytes (**C**,**D**). LRP-1 was detected by immunofluorescence staining and imaged by fluorescence microscopy (**G**,**H**), and quantitatively by immunoblotting in endothelial cells (**E**,**F**). Data was analyzed using one or two-way ANOVA analysis followed by Dunnett multiple comparisons. A value of *p* < 0.05 was considered significant. * vs. media.

**Figure 6 cells-12-00632-f006:**
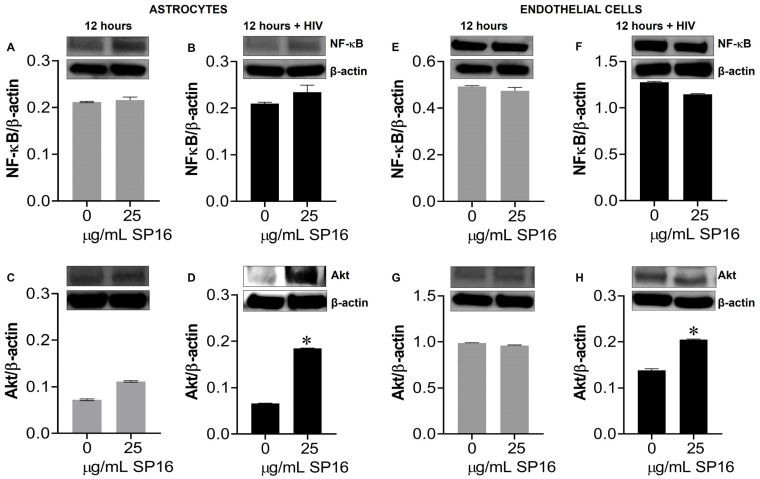
SP16 increases the expression levels of Akt in astrocytes and endothelial cells. Cell lysates were collected after 12 h and used to measure NF-κB (**A**,**B**,**E**,**F**) and total Akt (**C**,**D**,**G**,**H**) expression levels by immunoblotting. Gray bars = non-infected cells, black bars = infected cells. Results are reported as the mean ± SEM of three independent experiments. Data was analyzed using one or two-way ANOVA analysis followed by Dunnett multiple comparisons. A value of *p* < 0.05 was considered significant. * vs. media.

**Figure 7 cells-12-00632-f007:**
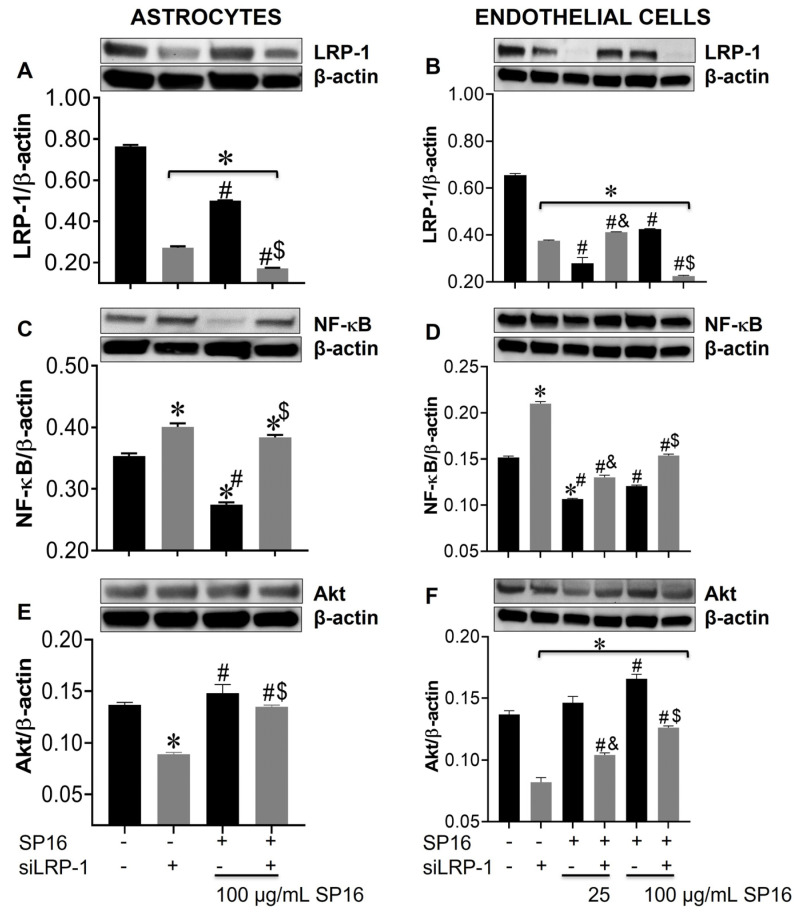
Silencing the *LRP-1* gene leads to increased expression levels of NF-κB and decreased levels of Akt in astrocytes and endothelial cells. Cell lysates from astrocytes (**A**,**C**,**E**) and endothelial cells (**B**,**D**,**F**) were collected after 12 h and LRP-1 (**A**,**B**), NF-κB (**C**,**D**), and Akt (**E**,**F**) expression were measured by immunoblotting. Results are reported as the mean ± SEM of three independent experiments. Data were analyzed using one or two-way ANOVA analysis followed by Dunnett multiple comparisons. A value of *p* < 0.05 was considered significant. * vs. SP16/-siLRP-1; ^&^ vs. 25 μg/mL SP16; ^$^ vs. 100 μg/mL SP16/-siLRP-1; ^#^ vs. SP16.

**Figure 8 cells-12-00632-f008:**
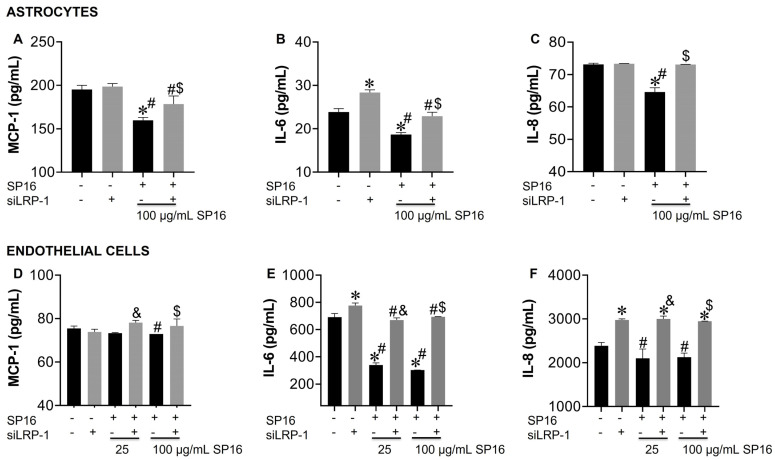
LRP-1 mediates the anti-inflammatory action of SP16 in astrocytes and endothelial cells. Supernatant from astrocytes (**A**–**C**) and endothelial cells (**D**–**F**) collected after 12 h was used to measure inflammatory molecules by ELISA. Results are reported as the mean ± SEM of three independent experiments. Data were analyzed using one or two-way ANOVA analysis followed by Dunnett multiple comparisons. A value of *p* < 0.05 was considered significant. * vs. SP16/-siLRP-1; ^&^ vs. 25 μg/mL SP16; ^$^ vs. 100 μg/mL SP16/-siLRP-1; ^#^ vs. SP16.

**Figure 9 cells-12-00632-f009:**
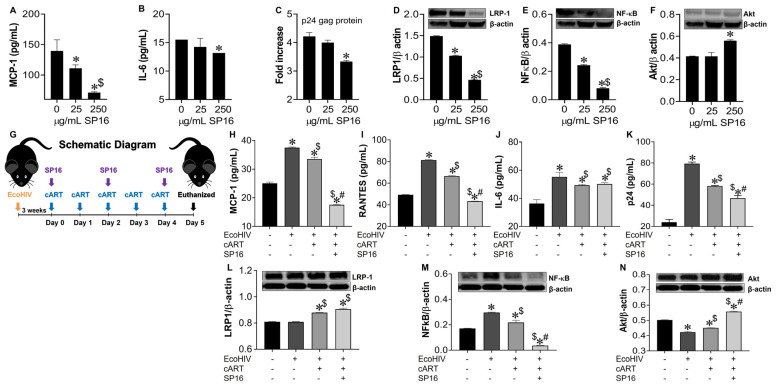
SP16 induces anti-inflammatory and anti-viral responses in EcoHIV-infected mouse brain. Supernatant was collected after 24 h post treatment and used to measure the secretion of inflammatory molecules (**A**,**B**) and viral titer (**C**) by ELISA. LRP-1 (**D**), NF-κB (**E**), and Akt (**F**) protein expression levels were measured by immunoblotting. Results are reported as the mean ± SEM of 3 independent experiments. Data were analyzed using one or two-way ANOVA analysis followed by Dunnett multiple comparisons. A value of *p* < 0.05 was considered significant. * vs. media; ^$^ vs. 25 μg/mL SP16. Brains of adult mice (N = 3–5/treatment) were collected at necropsy. EcoHIV Infected animals received cART alone and in combination with SP16 (as illustrated in schematic diagram **G**). Homogenized brain lysates were used to measure secretion of inflammatory molecules (**H**–**J**) and viral titer (**K**) by ELISA. LRP-1 (**L**), NF-κB (**M**), and Akt (**N**) protein expression levels were measured by immunoblotting. Data were analyzed using one or two-way ANOVA analysis followed by Dunnett multiple comparisons. A value of *p* < 0.05 was considered significant. * vs. PBS control; ^$^ vs. EcoHIV; # vs. cART.

## Data Availability

Data is contained within the article or Appendix A.

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
