# Peer review of "SERPIN-Derived Small Peptide (SP16) as a Potential Therapeutic Agent against HIV-Induced Inflammatory Molecules and Viral Replication in Cells of the Central Nervous System"

_cells, 2023, doi:10.3390/cells12040632_

Round 1
Reviewer 1 Report
In this interesting manuscript, Soler et al. describe the anti-inflammatory and anti-viral effect of SP16 as a potential therapeutic strategy for HIV-induced inflammation in the CNS. Although the experiments and the hypothesis of the manuscript are interesting, I have major concerns about western blot quantifications that need to be clarified, in addition to other suggestions/comments that I detail down below:
- In the Results section, I suggest to use the subheaders as conclusion, not the experiment that was done or the function that is measured (same for figure legends). For example, the first one (“Anti-inflammatory and anti-viral responses of SP16 in microglia…”) could be replaced with “SP16 induces anti-inflammatory response in microglia…” and so on. It would be better for readers and summarize the main concept for each section.
- Infected microglia does not respond differently compared to uninfected microglia, either with or without BBB. Could authors provide a control without virus in the p24 ELISA to determine that these cells are infected?
- In lines 258-260, authors mention they measured TNFa and RANTES, but results are not shown in Figure 1.
-Line 305 and Figure 2, please specify which panel has information of uninfected cells and which one has information of infected cells.
-It is unclear when authors explain in methods that cell viability was expressed as the percentage of live cells out of total cells, and then ranges of 5-15% viability in Figure 4. Does it mean that is % of live cells or % cell death?
- A proper gating strategy for flow cytometry must be shown in Figure 5, and with better resolution. In addition, an FMO is needed to determine what can be considered positive staining, it is possible that the peak excluded in 5A represents microglial cells expressing LRP1.
- The explanation given in the discussion about beta actin levels in the 250 ug condition in Fig 5 and 6 would be valid and reasonable, but we observe in Fig. 6C, D, G and H that beta actin levels are similar to the other conditions (lower concentrations of SP16). Please clarify. In addition, NFkB quantification does not represent what it’s shown in the representative picture in Fig. 6E, a stronger signal in NFkB compared to beta actin should show a bigger column compared to other concentrations. Please clarify how these quantifications were done and if they were relative to beta actin signal. Same happens in Fig. 7B, for example. Please chose carefully representative pictures from plots.
- Authors should rephrase lines 406-408, the role of LRP1 has been described in the literature, maybe specify the role of SP16 in LRP1 signaling, or similar.
- Authors measure total Akt levels but not phosphorylation of Akt, the statement in line 432 should be modified. Same statement is done in the discussion (line 570).
- Please, remove “Must be in Times roman” from Fig.9E.
-The experimental approach in Fig. 9A-F shouldn’t be considered ex vivo, because cells are following the same experimental design as in previous figures (in vitro). In addition, the schematic should be labeled as a specific panel, so it can be mentioned in line 466.
-For Supplemental Fig.2, please add scale bar and specify thickness of brain sections. Add scale bars or magnification also to other figures with pictures.
-In line 526, authors say that a higher concentration may attenuate IL6, IL8 and p24 release in Figure 1 E, G and E. Given that a 250 ug concentration was used in other figures, it is something that could have been checked.
- I suggest using a different terminology for molecule gradient or transport, and avoid the term migration or transmigration, that would be use for cells (line 529).
- Several typos have been found in the text, please read carefully and correct them.
-Add reference in line 561 to reference list.
- Supplemental fig.3 should be described in the Results section. No quantification of cell death or cell viability has been shown in this figure, so the pictures provided are not enough to support the potential neuroprotective role of SP16 in culture. Given that the manuscript if focused on glial cells/BBB and the lack of quantification, I would recommend to remove this figure from the manuscript. To support the hypothesis suggested in line 619, western blot analysis could be done, as in previous figures. Instead using lower concentration of FBS, conditioned media from supernatants of other experiments with astrocytes/microglia/endothelial cells could be used, and would model better this hypothesis.
- I would reword statements in line 630, tissues do not secrete these molecules but the cells, or something like “high levels of XXX were found in brain lysates”. Same in line 637, “… to induce cytokines” (cytokine secretion or similar)
Author Response
Reviewer 1
We would like to thank reviewer 1 for his/her insightful comments and have responded to each below and in blue font in the body of the manuscript. We were pleased with the overall outcome and enthusiasm by the reviewer and feel that the comments and suggested provided strengthened the manuscript.
Comment 1. In the Results section, I suggest to use the subheaders as conclusion, not the experiment that was done or the function that is measured (same for figure legends). For example, the first one (“Anti-inflammatory and anti-viral responses of SP16 in microglia…”) could be replaced with “SP16 induces anti-inflammatory response in microglia…” and so on. It would be better for readers and summarize the main concept for each section.
Response: We thank the reviewer for his/her comments and have made the changes to the sub-headers and figure legends, accordingly.
Comment 1. Infected microglia does not respond differently compared to uninfected microglia, either with or without BBB. Could authors provide a control without virus in the p24 ELISA to determine that these cells are infected?
Response: We have now included a control without virus, as an inset, in Figure 1.
Comment 3. In lines 258-260, authors mention they measured TNF-a and RANTES, but results are not shown in Figure 1.
Response: We thank the reviewer for his/her careful observation and a clear oversight on our part. Originally, the ELISA data included the cytokines TNF-alpha and RANTES. However, minor significances were detected as compared to the data representing MCP-1, IL-6, and IL-8. We have removed the lines mentioning TNF and RANTES in regard to Figure 1.
Comment 4. Line 305 and Figure 2, please specify which panel has information of uninfected cells and which one has information of infected cells.
Response: Figure 2 has been revised to specify which findings represents cells infected and non-infected with HIV.
Comment 5. It is unclear when authors explain in methods that cell viability was expressed as the percentage of live cells out of total cells, and then ranges of 5-15% viability in Figure 4. Does it mean that is % of live cells or % cell death?
Response: We thank the reviewer for his/her careful observation and a clear oversight on our part. The Y-axis for MTT represents OD and the Y-axis for Trypan blue represents % viability. Moreover, during revision we realized that the concentrations for astrocytes and endothelial cells, were mislabeled. We have revised Figure 4.
Comment 6. A proper gating strategy for flow cytometry must be shown in Figure 5, and with better resolution. In addition, an FMO is needed to determine what can be considered positive staining, it is possible that the peak excluded in 5A represents microglial cells expressing LRP1.
Response: We have revised Figure 5 and included a table representing the numbers corresponding to the gated cells expressing LRP-1.
Comment 7. The explanation given in the discussion about beta actin levels in the 250 ug condition in Fig 5 and 6 would be valid and reasonable, but we observe in Fig. 6C, D, G and H that beta actin levels are similar to the other conditions (lower concentrations of SP16). Please clarify.
Response: We thank the reviewer for his/her comment. We were confused as well, by some incongruities seen with b-actin, which could account for a number of reasons, including the ones mentioned in the discussion. However, after re-analyzing the findings for the human primary cells (astrocytes and endothelial), we concluded that the cells might be dying at 250 µg concentrations of SP16. We have revised Figures 5 and 6.
Comment 8. In addition, NFkB quantification does not represent what it’s shown in the representative picture in Fig. 6E, a stronger signal in NFkB compared to beta actin should show a bigger column compared to other concentrations. Please clarify how these quantifications were done and if they were relative to beta actin signal. Same happens in Fig. 7B, for example. Please chose carefully representative pictures from plots.
Response: We have re-calculated the images using the Image J Software and quantifications were relative to beta actin signal. We have revised Figures 6 and 7.
Comment 9. Authors should rephrase lines 406-408, the role of LRP1 has been described in the literature, maybe specify the role of SP16 in LRP1 signaling, or similar.
Response: We have rephrased the sentence and now reads as follows: Using gene silencing approach, we confirmed the role of LRP-1 in the regulation of NF-kB and Akt expression in astrocytes (Figure 7 A, C, E) and endothelial cells (Figure 7 B, D, F).
Comment 10. Authors measure total Akt levels but not phosphorylation of Akt, the statement in line 432 should be modified. Same statement is done in the discussion (line 570).
Response: We have modified the statement and indicated “total Akt” in the figure legend
Comment 11. Please, remove “Must be in Times roman” from Fig.9E.
Response: We have removed the phrase in Figure 9.
Comment 12. The experimental approach in Fig. 9A-F shouldn’t be considered ex vivo, because cells are following the same experimental design as in previous figures (in vitro). In addition, the schematic should be labeled as a specific panel, so it can be mentioned in line 466.
Response: We changed “ex vivo” to “in vitro”. Schematic diagram is now labeled as Figure 9G, in the figure and in the text.
Comment 13. For Supplemental Fig.2, please add scale bar and specify thickness of brain sections. Add scale bars or magnification also to other figures with pictures.
Response: We added a scale bar, and we measured the RFU in revised Supplemental Figure 2. We have specified the thickness (15 um) in the text.
Comment 14. In line 526, authors say that a higher concentration may attenuate IL6, IL8 and p24 release in Figure 1 E, G and E. Given that a 250 ug concentration was used in other figures, it is something that could have been checked.
Response: We have removed this statement because at a higher concentration we did not see significant differences. As suggested by the second reviewer, the lack of anti-inflammatory response by SP16 could be due to other factors including peptide modification during transportation across the BBB monolayer.
Comment 15. I suggest using a different terminology for molecule gradient or transport, and avoid the term migration or transmigration, that would be use for cells (line 529).
Response: We have replaced the terms migration and transmigration with “transportation” when referring to SP16 and we replaced transmigration with “permeability” when referring to FITC-dextran.
Comment 16. Several typos have been found in the text, please read carefully and correct them.
Response: We appreciate the comment. We have corrected the typos and we have reached out to a native English speaker, who has edited and revised the text. Changes are indicated in blue font.
Comment 17. Add reference in line 561 to reference list.
Response: The reference was removed from the text, since we are no longer showing in vitro data exposed to high concentration of SP16
Comment 18. Supplemental fig.3 should be described in the Results section. No quantification of cell death or cell viability has been shown in this figure, so the pictures provided are not enough to support the potential neuroprotective role of SP16 in culture. Given that the manuscript if focused on glial cells/BBB and the lack of quantification, I would recommend to remove this figure from the manuscript.
Response: We have removed Figure 3 from supplemental data.
Comment 19. I would reword statements in line 630, tissues do not secrete these molecules but the cells, or something like “high levels of XXX were found in brain lysates”. Same in line 637, “… to induce cytokines” (cytokine secretion or similar)
Response: We have rephrased the sentences. Changes are indicated in blue font in the text.

Reviewer 2 Report
This is an interesting study to determine the anti-inflammatory and potentially neuroprotective activity of serpin-derived SP16. While the study design and findings are overall compelling, there are some shortcomings that should be addressed.
1. Line 40 states that "cART is not able to transmigrate the BBB". This blanket statement is not correct as illustrated in Ferrara et al 2020 (PMID: 32694413).
2. Since inflammation, gliosis, and BBB are focus points of the study, it is necessary to include a paragraph in the background on what is known about these mechanisms in HIV+ human brain tissues and in rodent models for HIV- and ART-induced neurotoxicity.
3. The concentrations of SP16 should be included as mass/volume. Currently, only the mass of SP16 is listed in the methods, preventing reproduction of the experiments.
4. What was the justification for the cART regimen used in the mouse study?
5. what portion of the mouse brains were homogenized, whole brain?
6. Many errors throughout text, grammatical and spelling, for example: line 310: concertation?, Line 612: SerpinI1? There are many others; please give this a thorough reading.
7. The Results could be explained more clearly. For example, Figure 3: SP16-induced reductions are explained as 3-fold and 2-fold decreases, which don't look to be particularly accurate descriptions. A more succinct and clear way to explain would be do describe as % reductions compared to control.
8. In Figures 5 and 6, indeed LRP1 and NFkB are reduced, respectively, but in both blots, beta actin is also reduced proportionally. These are not particularly convincing blots. Is there another housekeeping gent that could be used?
9. Figure 7: LRP1 knockdown in astrocytes is not convincing. The graph shows a :75% reduction (7A), but the bands for both LRP1 and beta actin appear to be reduced in the blot. The knock down in endothelial cells looks convincing.
10. Why are different concentrations of SP16 used in different experiments (100 vs 250ug)?
11. Given the lack of SP16-induced reduction in cytokine expression in microglia in the BBB model (Figure 1), the authors astutely tested transmigration of sp16 across the BBB and speculated that although it is crossing, the amount crossing may be too low to have an effect on the microglia. Could it also be that during transmigration the SP16 is being modified by the BBB cells in a way that reduces its anti-inflammatory activity?
12. The Discussion is written much like the results. I'd suggest not restating the results, but discuss them in the context of previous studies with a similar paradigm. For example, discuss other studies of neuroprotection in rodent models for HIV-induced neurotoxicity, ie iTat, gp120tg mouse, HIV rat treated with rapamycin, cannabinoids, gene delivery, etc.
13. Clean up the text and make clear the language. For example. Line 641: "Despite the differences in viral genome our data showed similarities in findings". Similarities to what findings in what models?
Author Response
Reviewer 2:
We would like to thank reviewer 2 for his/her insightful comments and have responded to each below and in blue font in the body of the manuscript. We were pleased with the overall outcome and enthusiasm by the reviewer and feel that the comments and suggested provided strengthened the manuscript.
Comment 1. Line 40 states that "cART is not able to transmigrate the BBB". This blanket statement is not correct as illustrated in Ferrara et al 2020 (PMID: 32694413).
We have removed the statement and included a revised paragraph with a sentence that illustrates the postmortem findings cited by Ferrara et al., in lines 38-46.
Comment 2. Since inflammation, gliosis, and BBB are focus points of the study, it is necessary to include a paragraph in the background on what is known about these mechanisms in HIV+ human brain tissues and in rodent models for HIV- and ART-induced neurotoxicity.
We agree and we have added a new paragraph in lines 52-69 that reads as follows: HIV circulates throughout the bloodstream and enters the CNS during the first weeks of infection, mainly mediated by infected monocytes and CD4+ T lymphocytes. These cells are generally attracted to inflammation sites and can enter the perivascular spaces16. Monocytes are often described as the “Trojan Horse” in CNS viral entry17, 18. HIV can also reach the CNS through the lymphocytes, which are capable of harboring viruses that can reproduce in macrophages19. Furthermore, neuro-invasion may also happen by circulating virus crossing the BBB intracellularly through endothelial cells, especially when the permeability of this layer is compromised16. When the virus reaches the brain by either one of these pathways, it infects and activates microglia. These cells can also be activated by viral proteins such as gp120 and Tat released from infected cells. The activated microglia then release arachidonic acid, quinolinic acid, glutamate, L-cyteine, platelet-activating factor (PAF), and free radicals. These neurotoxic substances induce synaptic damage and neuronal injury and contribute to astrocyte activation. Activated astrocytes release glutamate, free radicals, and other neurotoxic substances that induce metalloproteinases, increase calcium influx, and further aggravate neuronal damage20. This creates an ideal environment for viral replication, chronic inflammation, neurotoxicity, and ultimately neurodegeneration21.
Comment 3.The concentrations of SP16 should be included as mass/volume. Currently, only the mass of SP16 is listed in the methods, preventing reproduction of the experiments.
We changed the concentration in the text and figures to µg/µL.
Comment 4.What was the justification for the cART regimen used in the mouse study?
The representative cART regimen was based on validations by others and is now cited in the text.
Comment 5.What portion of the mouse brains were homogenized, whole brain?
We added the following paragraph in the method section: After treatments (at day 5), mice were sacrificed and whole brains, excluding the brainstem and the olfactory bulb, were collected. Half of the recovered brain hemisphere was used for immunohistochemistry and LC-MS/MS while the other half was homogenized and used for biochemical assays. For protein biochemical assays, postmortem brains were homogenized in cell lysis buffer (Thermo Scientific, Waltham, MA, USA) using a bead-beater apparatus (MagNA Lyser, Roche, Indianapolis).
Comment 6.Many errors throughout text, grammatical and spelling, for example: line 310: concertation?, Line 612: SerpinI1? There are many others; please give this a thorough reading.
The manuscript has been edited by a native speaker.
Comment 7. The Results could be explained more clearly. For example, Figure 3: SP16-induced reductions are explained as 3-fold and 2-fold decreases, which don't look to be particularly accurate descriptions. A more succinct and clear way to explain would be do describe as % reductions compared to control.
Results are described as % change.
Comment 8. In Figures 5 and 6, indeed LRP1 and NFkB are reduced, respectively, but in both blots, beta actin is also reduced proportionally. These are not particularly convincing blots. Is there another housekeeping gent that could be used?
We thank the reviewer for this observation. After re-analyzing the data, we believe that the cells might be dying at 250 µg concentrations. We have removed the findings at 250 ug SP16 and revised the images in Figures 5 and 6.
Comment 9. Figure 7: LRP1 knockdown in astrocytes is not convincing. The graph shows a :75% reduction (7A), but the bands for both LRP1 and beta actin appear to be reduced in the blot. The knock down in endothelial cells looks convincing.
We have revised Figure 7.
Comment 10. Why are different concentrations of SP16 used in different experiments (100 vs 250ug)?
We used different concentrations of SP16, to measure a concentration-dependent response of the peptide. The maximum concentration used at the beginning of the experiment was 150 ug, however, in later experiments we realized that higher concentration of 250 ug would elicit a stronger response, especially in experiments using murine glial culture. Although this was not the case in experiments using primary human cells.
Comment 11. Given the lack of SP16-induced reduction in cytokine expression in microglia in the BBB model (Figure 1), the authors astutely tested transmigration of SP16 across the BBB and speculated that although it is crossing, the amount crossing may be too low to influence the microglia. Could it also be that during transmigration the SP16 is being modified by the BBB cells in a way that reduces its anti-inflammatory activity?
We thank the reviewer for his/her suggestion. At this point we can only speculate about the faith of SP16 once it crosses the BBB. As suggested by the reviewer, peptide modification by the BBB could be a possibility. In general, movement of molecules across the endothelium can occur through bulk-phase transport or the more selective process of receptor-mediated endocytosis and favors apical to basolateral transport because of the concentration gradient on the blood side of the endothelium [PMID: 12843411]. In the case of the serine protease inhibitor alpha 1 anti-trypsin, in the lungs, it was shown to be taken up primarily by clathrin, and caveolae-dependent receptor endocytosis, both of which have been implicated in transcytosis of molecules across the endothelium [PMID: 19423638]. Furthermore, clathrin-dependent receptor uptake leads to transcytosis while caveolae-dependent receptor uptake leads to degradation [PMID: 12843411]. However, to monitor SP16 trafficking across the BBB in real time, requires additional techniques, beyond the scope of this manuscript.
Comment 12. The Discussion is written much like the results. I'd suggest not restating the results, but discuss them in the context of previous studies with a similar paradigm. For example, discuss other studies of neuroprotection in rodent models for HIV-induced neurotoxicity, ie iTat, gp120tg mouse, HIV rat treated with rapamycin, cannabinoids, gene delivery, etc.
We have revised the Discussion and included findings from previous studies. Revision are indicated in blue font.
Comment 13. Clean up the text and make clear the language. For example. Line 641: "Despite the differences in viral genome our data showed similarities in findings". Similarities to what findings in what models?
We have revised the text, moreover we have asked a native speaker to edit the manuscript.

Round 2
Reviewer 1 Report
All my comments have been addressed by the authors. I'd like to wish the authors good luck in future research.
Reviewer 2 Report
The authors have addressed all concerns and the manuscript is significantly improved compared to the former version.